# Normalizing Flows for Calibration and Recalibration

## Abstract

In machine learning, due to model misspecification and overfitting, estimates of the aleatoric uncertainty are often inaccurate. One approach to fix this is isotonic regression, in which a monotonic function is fit on a validation set to map the model's CDF to an optimally calibrated CDF. However, this makes it infeasible to compute additional statistics of interest on the model distribution (such as the mean). In this paper, through a reframing of recalibration as MLE, we replace isotonic regression with normalizing flows. This allows us to retain the ability to compute the statistical properties of the model (such as closed-form likelihoods, mean, correlation, etc.) and provides an opportunity for additional capacity at the cost of possible overfitting. Most importantly, the fundamental properties of normalizing flows allow us to generalize recalibration to conditional and multivariate distributions. To aid in detecting miscalibration and measuring our success at fixing it, we use a simple extension of the calibration Q-Q plot.

## 1 Introduction

Recent advances in deep learning have led to models with significantly higher overall accuracy on both classification and regression tasks compared to what was achievable in the past. However, an important component in conjunction with accuracy is a model's ability to accurately assess the uncertainty in its prediction. Most taxonomies classify uncertainty into three sources: approximation, aleatoric, and epistemic uncertainty (Der Kiureghian & Ditlevsen, 2009). Approximation uncertainty quantifies the error from fitting a simple model to complex data. Aleatoric uncertainty quantifies the uncertainty of the conditional distribution of the target variable given features. This uncertainty arises from hidden variables or measurement errors and cannot be reduced through collecting more data under the same experimental conditions. Epistemic uncertainty quantifies the uncertainty arising from fitting a model utilizing finite data, i.e. it is inversely proportional to the density of the training examples and can be reduced by collecting data in the low density regions.

These different sources of uncertainty have different techniques for handling them. Using high capacity models such as neural networks removes a large part of the approximation uncertainty. By fitting a full distribution on the target conditional on features, we can model the aleatoric uncertainty from observations. Inaccurate estimates of aleatoric uncertainty can be explained by underfitting (insufficient complexity in the conditional distributions) or overfitting (models with sufficient capacity can memorize the data, leading to the distributions collapsing to deltas). Though epistemic uncertainty is important for the model to answer what it does not know, the focus of this paper is on improving estimates of the aleatoric uncertainty.

Our approach in this paper is to handle both model fit and calibration using normalizing flows. Normalizing flows can be used in conjunction with amortized inference to improve the flexibility of the output distribution, and further, through a reframing of recalibration as maximum likelihood estimation (MLE), normalizing flows can be used to handle any miscalibration found on a validation set. Further, we use a simple extension of the calibration plot from Kuleshov et al. (2018) to help with the the analysis of the calibration of a model across different regions of the data.

## 2 RELATED WORK

One method for handling aleatoric uncertainty is amortized inference with Gaussians (Lakshminarayanan et al., 2017; Nix & Weigend, 1994; Kendall & Gal, 2017) where a model, such as a neural network, maps from features to the parameters of a Gaussian. This approach models aleatoric uncertainty directly but suffers from approximation uncertainty as a Gaussian cannot model complex targets.

Another approach is Bayesian methods such as Bayesian Ridge Regression (Tipping, 2001) and MC Dropout (Gal & Ghahramani, 2016). Similar to amortized inference with Gaussians, the output distribution limits the capacity of the model. Full Bayesian techniques with neural networks are often too computationally expensive in practice, and approximate methods often fail to capture the full complexity of the uncertainty (Lakshminarayanan et al., 2017).

Another family of methods uses quantile regression with non-linear techniques such as decision trees or neural networks. Some of these methods require a predefined set of quantiles (Takeuchi et al., 2006; Wen et al., 2017; Rodrigues & Pereira, 2018; Taylor, 2000). Simultaneous Quantile Regression (SQR, Tagasovska & Lopez-Paz (2019)) trains one model on all quantiles and is able to learn complex shaped distributions. However the training procedure requires the model to learn to be monotonic instead of being constrained to be so and is not trivial to extend to multidimensional outputs. Pearce et al. (2018) learns a finite set of quantiles by using quality metrics for predictive intervals.

Normalizing flows have been used in the contexts of variational inference and generative modeling. The approaches to normalizing flows can be categorized into autoregressive methods (Kingma et al., 2016; Papamakarios et al., 2017; Huang et al., 2018; Cao et al., 2019), coupling layers (Dinh et al., 2014; 2016; Kingma & Dhariwal, 2018; Ho et al., 2019), residual networks (Rezende & Mohamed, 2015; van den Berg et al., 2018; Gopal, 2020) and continuous flows (Grathwohl et al., 2018).

## 3 BACKGROUND

### 3.1 RECALIBRATION

An important goal in modeling is to have well-calibrated distributions as this allows users of the model to understand the confidence the model places on its prediction. In other words, with a well-calibrated model, we can better ascertain the uncertainty in the model's predictions and respond differently to those predictions depending on the uncertainty surrounding them.

Guo et al. (2017) showed that unlike techniques used decades ago such as Bayesian Ridge Regression, modern neural network-based classifiers are very poorly calibrated. A simple variant of Platt scaling and other histogram based techniques applied to a validation set were shown to help alleviate the calibration problem where perfect calibration is defined as

$$\mathbb{P}(\hat{Y} = Y | \hat{P} = p) = p, \quad \forall p \in [0, 1]$$

where $\hat{Y}$ is a class prediction and $\hat{P}$ is its predicted probability of correctness.

Kuleshov et al. (2018) extended the analysis in Guo et al. (2017) to neural network-based regressors; isotonic regression, a method for learning monotonic univariate functions, was applied to map from $\mathcal{F}_{x_j}(y_j) = \hat{p}_j$ to $|\{y_n | \mathcal{F}_{x_n}(y_n) < \hat{p}_j, n = 1, \ldots, N\}|/N$ (the fraction of the data where the model CDF is less than $\hat{p}$) to improve calibration where perfect calibration is defined as

$$\mathbb{P}(Y < \mathcal{F}_X^{-1}(p)) = p, \quad \forall p \in [0, 1]$$

where $\mathcal{F}_X$ is the predicted CDF function.

Kuleshov et al. (2018) further introduced calibration error as a metric to quantitatively measure how well the quantiles are aligned:

$$\hat{p}_j = |\{y_n | \mathcal{F}_{x_n}(y_n) < p_j, n = 1, \ldots, N\}| / N$$

$$\text{cal}(y_1, \ldots, y_N) = \sum_{j=1}^{M} (p_j - \hat{p}_j)^2 \tag{1}$$

where $M$ is the number of quantiles that are evaluated. In this paper, we set this to 100 evenly spaced quantiles.

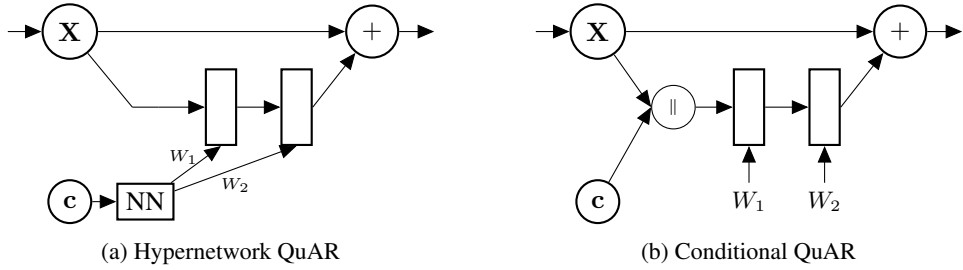

(a) Hypernetwork QuAR  (b) Conditional QuAR

Figure 1: A comparison between using a hypernetwork approach versus conditional approach to creating Conditional QuAR Flows to model $p(X|c)$ where $W_1$ and $W_2$ parametrize the layers in the residual connection.

## 3.2 Normalizing Flows

A crucial component in improving uncertainty estimates in this paper is to fit normalizing flows, a generative model for density estimation using invertible functions.

Suppose that we wish to formulate a joint distribution on an $n$-dimensional real vector $x$. A flow-based approach treats $x$ as the result of a transformation $g$ applied to an underlying vector $z$ sampled from a base distribution $p_z(z)$. The generative process for flows is defined as:

$$z \sim p_z(z)$$
$$x = g(z)$$

where $p_z$ is often a Normal distribution and $g$ is an invertible function. Notationally, we will use $f = g^{-1}$. Using change of variables, the log likelihood of $x$ is

$$\log p_x(x) = \log p_z\left(f(x)\right) + \log\left|\det\left(\frac{\partial f(x)}{\partial x}\right)\right|$$

To train flows (i.e. maximize the log likelihood of data points), we need to be able to compute the logarithm of the absolute value of the determinant of the Jacobian of $f$, also called the *log-determinant*. To construct large normalizing flows, we can compose smaller ones as this is still invertible and the log-determinant of this composition is the sum of the individual log-determinants.

An important observation is that every univariate distribution can be viewed as a flow in which the base distribution $p_z(z)$ is a Uniform distribution over $[0, 1]$ and $g$ is the inverse CDF of the univariate distribution. This can be generalized to multivariate distributions using the chain rule.

## 3.3 Conditional QuAR Flows

In this paper, the specific normalizing flow used is Quasi-Autoregressive Residual Flows (QuAR Flows, Gopal (2020)). QuAR Flows were shown to handle modeling complex distributions while having nice optimization properties and computationally efficient log-likelihoods. Other commonly used normalizing flows have drawbacks. Coupling layers do not work for one-dimensional inputs. Autoregressive flows are often conditional Gaussians which means in the one-dimensional case, the flow is simply a Gaussian. Residual Flows (Chen et al., 2019) are expressive but, though equivalent to QuAR Flows in the one-dimensional case, are computationally expensive for high-dimensional distributions.

If we would like to condition a QuAR Flow given some features $c$, we could use a hypernetwork that maps $c$ to the parameters of the QuAR Flow (Figure 1a). However, this can be computationally expensive if the QuAR Flow is parameterized with thousands of parameters. Instead, conditional information can be incorporated by concatenating the conditions $c$ before applying the first layer in the residual connection within a QuAR Flow (Figure 1b).

## 4 REGRESSION USING NORMALIZING FLOWS

The goal of regression is to model the conditional probability distribution $p(y|x)$. Frequently, this task is reduced to obtaining point estimates of the mean or median by minimizing the mean squared error or mean absolute error respectively. In order to access uncertainties, we must model the full conditional distribution; in addition to this, we can still retrieve point estimates of the mean, median, or any other distributional statistic. In this section, we discuss how normalizing flows can be used to this end as well as how recalibration can be reframed as maximum likelihood estimation (MLE).

### 4.1 CALIBRATION WITH NORMALIZING FLOWS

One approach to modeling aleatoric uncertainty is to use amortized inference with a Normal distribution as its output distribution, conditioned on features (Lakshminarayanan et al., 2017). However, if a Gaussian is not appropriate to fit the target, then the error in miscalibration can be attributed to approximation error, i.e. insufficient capacity to learn the distribution. A simple example of this is where we have no features to condition on (or all the features we have contain no relevant information) with a one-dimensional target. In Figure 2, we can see that since the target is bimodal, the Gaussian fit does not well describe the data.

To improve this model, we replace the Gaussian with a normalizing flow conditioned on our data, similar to Figure 1b. In this way, the distribution fit to the targets has sufficient capacity to learn complex distributions, including multimodal distributions (Figure 2).

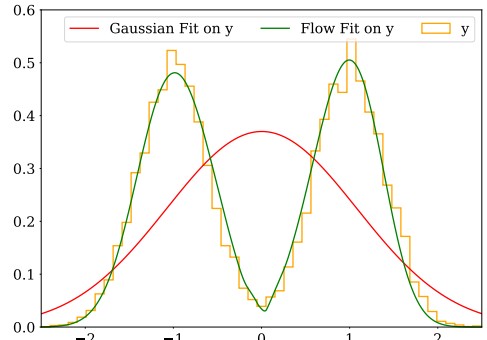

### 4.2 RECALIBRATION WITH NORMALIZING FLOWS

To handle overfitting of the uncertainty, we recalibrate the results of the original model where we assume the original model outputs a full distribution that has closed-form likelihoods. Since the only requirement is a full distribution with a closed-form likelihood, the original model that we recalibrate can be as simple as a Bayesian Linear Regression model

Figure 2: One-dimensional example of Gaussian versus Flow

or as complicated as a conditional normalizing flow. We recalibrate the original model by fitting a normalizing flow on top of it. In this way, the training process is the following two-step procedure:

1. $f_\theta = \arg\max_{f_\theta} \sum_{i=1}^{N_{tr}} \log p_{f_\theta}(y_i^{tr}|x_i^{tr})$
2. $g_\phi = \arg\max_{g_\phi} \sum_{i=1}^{N_{val}} \log p_{g_\phi}(F_\theta(y_i^{val}|x_i^{val}))$

where $f_\theta(y|x)$ is the predicted pdf function given $x$, $F_\theta(y|x)$ is the CDF of $f_\theta(y|x)$, $\{x^{tr}, y^{tr}\}_{i=1}^{N_{tr}}$ is the training set and $\{x^{val}, y^{val}\}_{i=1}^{N_{val}}$ is the validation set. Though the formulation of recalibration above shows we can use any likelihood model for $g_\phi$, since isotonic regression is a method to fit a free form function to the CDFs, we choose to use normalizing flows, specifically QuAR Flows, as these are more flexible than using autoregressive flows such as MAFs (Papamakarios et al., 2017).

After the two-step training, the new generative process is:

$$z \sim \text{Uniform}(0, 1)$$
$$u = G_\phi^{-1}(z)$$
$$y = F_\theta^{-1}(u|x)$$

where $G_\phi$ is the CDF of $g_\phi$. The density function of the above generative process is simply the product of the individual density functions: $f_\theta$ and $g_\phi$. Given this view of the generative process, one interpretation of the two-step process is to first fit $\theta$ on the training set where $\phi$ is initialized so that $G_\phi^{-1}$ is the identity function, and then to fit $\phi$ on the validation set while keeping $\theta$ frozen. Though $\theta$

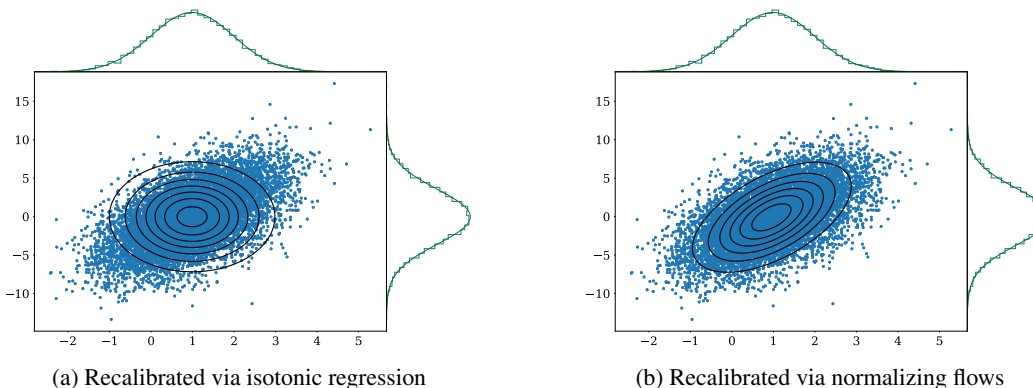

(a) Recalibrated via isotonic regression  (b) Recalibrated via normalizing flows

Figure 3: Comparison plots of isotonic regression and flow recalibration on multiple outputs. The two axes are different targets being modeled, and the scatter plot shows that the targets are correlated. The top and right sides show the marginal histogram as well as the fit from the recalibrated distribution. The contour plot shows the two-dimensional contours of the recalibrated distribution which can be seen to not align well for isotonic regression.

and $\phi$ could be trained end-to-end, similar to the approach in Section 4.1, we show in our experiments (Section 6.2) that there are benefits to this two-step approach.

Given this log-likelihood viewpoint of recalibration, we can trivially incorporate conditioning of the normalizing flow on additional information. For example, say $f_\theta$ outputs $\mu$ and $\sigma$ that parameterize a Normal distribution, we could then condition $g_\phi$ on these two outputs. If we were to use many features to condition on, we would return to the first situation where we could overfit to the finite number of observations we have. We can think of the first model $f_\theta$ as compressing/embedding the data into two dimensions, and the second model utilizing the embedding to create a meaningful and well-calibrated distribution on the target. Compared to isotonic regression, we can easily incorporate conditional information while retaining monotonicity as normalizing flows are monotonic by construction.

Recalibration is interesting in that it is useful for both overfitting as well as underfitting. From Guo et al. (2017), we see that multinomial classification models have sufficient capacity to overfit uncertainties and benefit from recalibration. On the other hand, the simple example in Figure 2 suffers from miscalibrated uncertainties due to approximation uncertainty (underfitting), and recalibration reduces the approximation error.

One solution to handling overfitting is to early-stop on a validation set. Whether early stopping occurs or not, our solution incorporates the information of the validation set to correct potential overfitting. However, if we try to correct for multiple sources of overfitting, then we could introduce a new overfitting problem. In essence, there is a tradeoff between the validation size (which in turn controls training size) and how many sources of overfitting one can try to correct after training.

### 4.3 MULTIVARIATE RECALIBRATION

Whereas Platt scaling (Guo et al., 2017) and isotonic regression (Kuleshov et al., 2018) have been shown to improve the calibration of distributions when we have one-dimensional outputs, scaling these solutions to high-dimensional targets is non-trivial.

Say we are modeling two-dimensional real-valued targets and use isotonic regression to calibrate the targets individually. For simplicity, say we use amortized inference with independent output distributions. If the targets are not conditionally independent, the distribution will not be well-calibrated, even after recalibration with isotonic regression (Figure 3a). However, using normalizing flows, we can compute the CDF of the observed targets per output distribution and recalibrate this two-dimensional distribution (Figure 3b).

The general procedure of recalibrating $N$-dimensional distributions is to fit a normalizing flow on the $N$-dimensional vector $x_i = F(y_i | y_{<i})$.

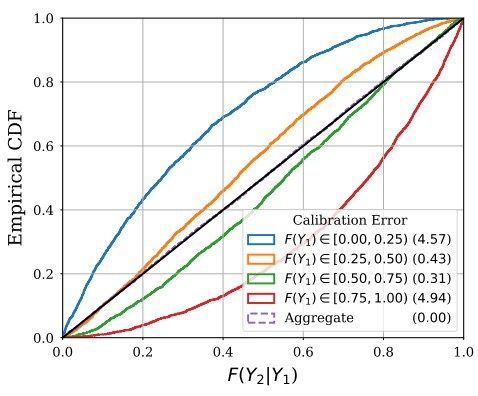 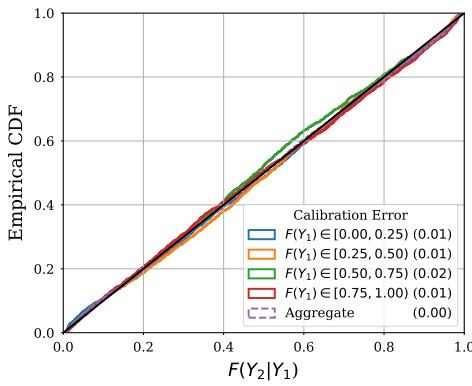

(a) Recalibration via isotonic regression        (b) Recalibration via normalizing flows

Figure 4: Calibration Q-Q plots for recalibration using isotonic regression and recalibration using flows for the simulated multivariate case in Figure 3a. The legend contains the calibration error for each curve.

## 5 MULTIVARIATE RECALIBRATION EXAMPLE

In multiclass classification, we often focus on individual metrics such as accuracy, precision, recall, F1, etc. However, exploring the accuracy of the separate classes using a confusion matrix can show if the model is biased even if the performance on average is good. Similar to this view for classification, we extend the calibration plot introduced in (Kuleshov et al., 2018) to show calibration performance on different subsets of the data, binned by either a feature or a prediction, e.g. Figure 4. The only restriction is the binning cannot be a function of the target we are evaluating.

Say we are modeling two-dimensional real-valued targets $(Y_1, Y_2)$ shown in Figure 3a. If we fit independent Gaussians to the target, the resulting distribution will not align with the contours of the data.

We first recalibrate the distribution by fittting isotonic regression models on $F(Y_1)$ and $F(Y_2|Y_1)$. The results for $F(Y_2|Y_1)$ are shown in Figure 4a where we can see that, though across the full dataset the calibration error is zero, when we bucket by $F(Y_1)$, the calibration error achieved is both significant and a strong function of $Y_1$ as it is unable to account for correlations in $Y_1$ and $Y_2$. In other words, the model is not calibrated in any given region of the data but only in aggregate.

However, if we recalibrate using normalizing flows by fitting a flow on the pair $(F(Y_1), F(Y_2|Y_1))$, we can see in Figure 4b that even if we bucket by $F(Y_1)$, the calibration error stays close to zero. Unlike isotonic regression, the flow can learn the correlation between the CDFs by being trained jointly on all of them. In addition to this, had we used a more complex calibration model that could model the correlations directly, we would not be able to compute the correlation after recalibrating with isotonic regression. By recalibrating with flows, we obtain a result that is still a distribution with a closed-form likelihood; this allows us to still be able to use the model to compute the correlation and any other distributional statistics.

In Figure 5, we show visually the difference between recalibration with flows and with isotonic regression. The key differences being that flows can recalibrate the CDFs jointly and be conditioned on additional information $c$, while isotonic regression is fit on each CDF independently and is unable to incorporate any additional information. The reason is that flows are monotonic by construction and can therefore handle multiple variables and conditions.

## 6 EXPERIMENTS

In this section, we analyze the performance of flows on multiple UCI datasets. Importantly, we do not aim for state-of-the-art performance, nor do we expect normalizing flows to outperform isotonic

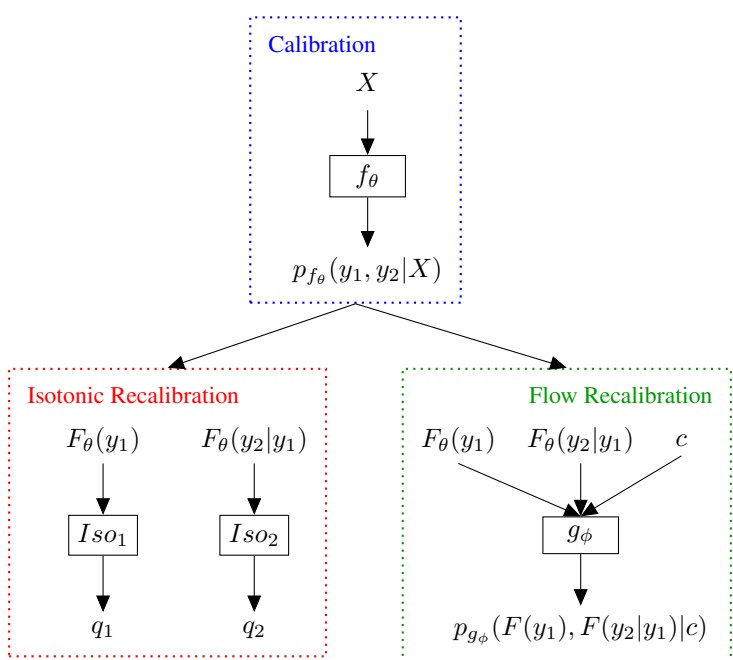

Figure 5: A diagram for multivariate recalibration using the same notation as that used in Section 4.2. The first step for recalibration is always the calibration step, shown in the blue box, where any likelihood model $f_\theta$ can be used. On the left, isotonic regression is used to recalibrate the CDFs from $f_\theta$; on the right, normalizing flows $g_\phi$ are used to recalibrate the CDFs from $f_\theta$ conditioned on some $c$.

regression in aggregate. In terms of calibration, we show that the increase in expressivity can be beneficial; in terms of recalibration, we show specific failure points of isotonic regression that binning uncovers and how normalizing flows fixes these issues.

## 6.1 CALIBRATION

To test the effect of increased capacity in distributions, we tested how well normalizing flows improve performance in distributional estimates compared to a few other popular approaches for estimating uncertainties. We do not compare directly against other methods for conditional distributions such as autoregressive flows for the reason specified in Section 3.3: in the one dimensional case, those are often the same as ConditionalGaussian. Instead of replacing the output distribution with a complex distribution created by hand, such as an explicit mixture, we rely on using more expressive flows which can model mixture behavior as well as more complicated behavior such as skew and kurtosis.

Table 1: Results before and after recalibration on the UCI dataset "naval-propulsion-plant". Details on the model methodology as well as performance on other UCI datasets are in Appendix B. The mean squared error (MSE) and its uncertainty is computed using 20 seeds. "Calib" is the calibration error (Equation 1) using 100 quantiles across 20 seeds instead of per seed.

|  | Before Recalib. | | Isotonic Recalib. | | Flow Recalib. | |
|---|---|---|---|---|---|---|
|  | MSE | Calib | MSE | Calib | MSE | Calib |
| ConditionalFlow | $0.009 \pm 0.004$ | 0.45 | N.A. | $< 0.01$ | $0.016 \pm 0.012$ | $< 0.01$ |
| ConditionalGaussian | $0.012 \pm 0.006$ | 2.35 | N.A. | $< 0.01$ | $0.012 \pm 0.006$ | $< 0.01$ |
| BayesianRidgeRegression | $0.055 \pm 0.004$ | 8.03 | N.A. | 0.01 | $0.48 \pm 0.02$ | $< 0.01$ |
| MC Dropout | $0.005 \pm 0.002$ | 0.51 | N.A. | N.A. | N.A. | N.A. |
| SQR | $0.0052 \pm 0.0012$ | 1.44 | N.A. | N.A. | N.A. | N.A. |

All results were evaluated on a held out test set that was not used for calibration or recalibration. Due to limited space, the full extent of the results and details of the models are left for Appendix A and Appendix B. To show the usefulness of the plot we introduced in Section 5, we focus on the results on the UCI dataset "naval-propulsion-plant" (Dheeru & Karra Taniskidou, 2017) in Table 1 and Section 6.2.

The models with consistent performance across UCI datasets in terms of calibration are Bayesian Ridge Regression, Conditional Gaussian, and Conditional Flow, with Conditional Flow often performing well in cases where Conditional Gaussian is unable to. Whereas these three models are well calibrated, the mean performance is much weaker for Bayesian Ridge Regression due to the limitations of linear functions.

## 6.2 RECALIBRATION

Of the models we tested, we recalibrated the Bayesian Ridge Regression, Conditional Gaussian, and Conditional Flow models. MC Dropout and SQR could also be recalibrated, though we chose not to as they lack closed form CDFs, meaning the technique would be approximate. In Table 1, we see the calibration errors after recalibrating with normalizing flows are comparable to those from isotonic regression. However, after isotonic regression is applied, only the calibration error can be computed. In contrast, we can still compute ordinary distribution statistics such as the mean after applying normalizing flows. Due to this property, we can even detect cases in which an improvement in calibration error coincides with a degradation of mean performance, suggesting if the goal were to *only* get point estimates of the mean, recalibration could be detrimental. For example, in Table 1 this was the case for Bayesian Ridge Regression.

In Figure 6a, whereas the calibration error metric shows the flow recalibrated model is well-calibrated averaged across the full dataset, we can see that see that, when bucketing by the predicted mean, the calibration error is significant.

To improve this, we recalibrate with a normalizing flow conditioned on the mean predicted by the original model. In this way, we see in Figure 6b that though the aggregate performance does not change, the performance is improved across different buckets. In essence we see that if we were analyzing the distribution across the dataset, we would be accurate; if we started to look closer to individual distributions, the performance is worse suggesting the quantiles of individual distributions cannot be trusted.

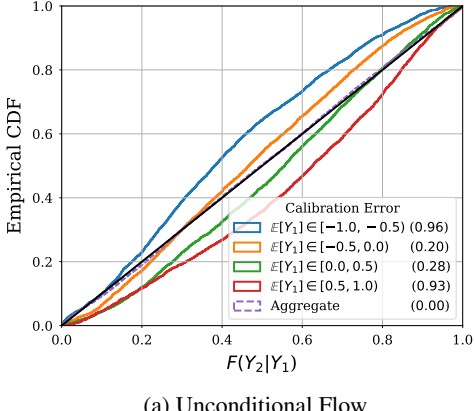
(a) Unconditional Flow

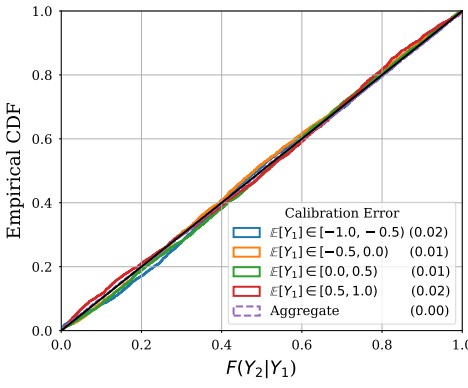
(b) Flow, Conditioned on Predicted Mean

Figure 6: Calibration Q-Q plots for recalibration with an unconditional flow and recalibration with a conditional flow on the dataset "naval-propulsion-plant". The legend contains the calibration error for each curve.

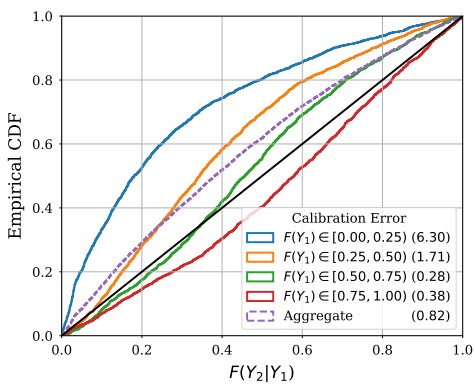
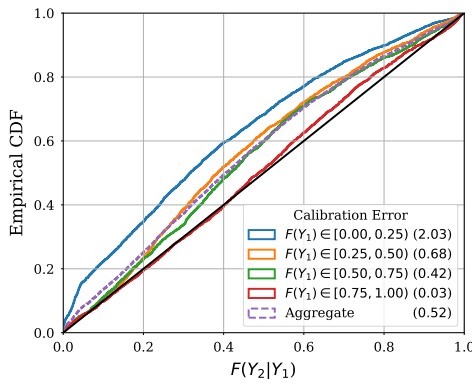

(a) Recalibrated via isotonic regression

(b) Recalibrated via normalizing flows

Figure 7: Calibration Q-Q plots for recalibration with isotonic regression and recalibration with normalizing flows on the second time step on the dataset "electricity". The legend contains the calibration error for each curve.

## 6.3 TIME SERIES

For our time series example, we trained a model on the UCI Dataset "electricity". We predict two time steps and show that even though we used an autoregressive model for calibration (the model could learn correlated distributions), there is still some correlation not captured that recalibration can help with.

Figure 7a shows that the calibration error of $F(Y_2|Y_1)$ when recalibrated independently with isotonic regression is significant. Figure 7b shows that a flow trained jointly on the CDFs significantly reduces the errors. Although the calibration error of the marginal distribution for the second step is comparable between the two approaches, the conditional performance favors normalizing flows. A possible explanantion of the still imperfect calibration of the flow model, though better than isotonic regression, is distribution shift.

## 7 CONCLUSION AND FUTURE WORK

In this paper, we addressed the calibration issue in modern neural networks through using higher capacity distributions via normalizing flows. We showed that normalizing flows can not only help during calibration, but also in recalibrating miscalibrated distributions, even improving the fit in the underfitting case. Though isotonic regression also uses a two-step producedure, we reframe the problem as MLE through which we can trivially generalize recalibration to the multivariate case. Though isotonic regression could be extended such that we could compute the log likelihood implied by its change of variables, normalizing flows are a more natural way to handle this with more research backing this technique, and are easily extendable to conditional and joint distributions. Interestingly, though the Conditional Flow has all it needs to be well calibrated, recalibration helps the overall calibration which was similarly found for classification in Guo et al. (2017); it remains for future work to better understand why and when this is useful.

Further, we extend the plot introduced in Kuleshov et al. (2018) by showing the calibration error on different subsets of the data, for example bucketed by mean prediction. Though this plot is able to show the calibration error across buckets, for future work, we will explore ways to visually show any trends in the miscalibration as well as whether the misalignments of the quantiles are statistically significant.

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

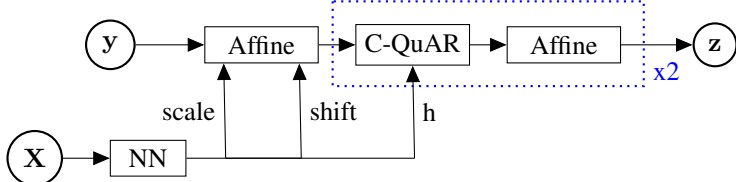

Figure 8: In this figure, we show our conditional flow model. A neural network takes $X$ and outputs the scale and shift of an affine flow and a condition vector for QuAR Flows. The condition used in the QuAR Flow is the same for both QuAR Flows. After the first conditional affine flow, there are two QuAR Flows with unconditional affine flows placed in between these two and at the end. The Conditional Flow model is equivalent to a Conditional Gaussian if we removed the QuAR Flows.

## A   MODEL TRAINING DETAILS

### A.1   CALIBRATION

We compare the performance of our model and multiple other popular approaches on 8 UCI datasets (Dheeru & Karra Taniskidou, 2017): boston-housing, concrete, energy, kin8nm, naval-propulsion-plant, power-plant, wine-quality-red and yacht. The models we compared Conditional Flows to were: Conditional Gaussian (amortized inference with Gaussians, Lakshminarayanan et al. (2017)), Bayesian Ridge Regression (Tipping, 2001), MC Dropout (Gal & Ghahramani, 2016), Simultaneous Quantile Regression (SQR) (Tagasovska & Lopez-Paz, 2019), Conditional Quantile (SQR where we only learn prespecified quantiles), Quality Driven (Pearce et al., 2018), and a Gradient Boosted Decision Tree with LightGBM (Ke et al., 2017).

Figure 8 shows a representation of the Conditional Flow network we used. The QuAR Flows are conditioned on a 64-dimensional representation from the condition network. Each Conditional QuAR Flow has two hidden layers in the residual connection with a hidden size 64 with ELU (Clevert et al., 2015) as our activation. The condition network has a single hidden layer with size 64 and ReLU as our activation. The QuAR Flows are conditioned on the 64-dimensional representation from the condition network.

For the other neural network approaches, we used networks with two hidden layers with size 128 and used ReLU as our nonlinearity. For hyperparameter tuning on all our neural networks, we searched the following grid:

- Learning rate: 1e-2, 1e-3, 1e-4
- Weight decay: 0, 1e-3, 1e-2, 1e-1
- Dropout Rates: 0.1 0.25, 0.5 0.75

The hyperparameters we used were taken from Tagasovska & Lopez-Paz (2019). We trained each of our neural networks for 2000 epochs with batch size 128.

For our LightGBM tree model (Ke et al., 2017), we used mean squared error as our loss to learn the mean of the output distribution. The grid of hyperparameters we searched was:

- Number of Trees: 100, 1000
- Maximum Depths: 3, 5, 7, 9
- Learning Rates: 1e-1, 1e-2, 1e-3, 1e-4
- Minimum Child Samples: 1, 20
- Dropout Rates: 0.1 0.25, 0.5 0.75

To make the tree model distributional, assuming homoscedasticity, we computed the standard deviation of the predicted normal distribution by computing the standard deviation of the residuals (observations minus predictions).

For SQR and MC Dropout, we computed the estimated mean and CDF from 1000 samples. For Conditional Flows, we computed the estimated mean by using the Gauss-Legendre quadrature with 100 points to approximate the integral.

## A.2 RECALIBRATION

For recalibration, we trained the recalibration model on the validation set of each of the 20 seeds. For the flow architecture, we stacked 8 QuAR Flows with affine flows in between each one as well as at the beginning and end. Since the input to the flow is a CDF $\in [0, 1]$, we also transform the data with a Logit Function as the first layer of the flow and account for the change of variables in the loss. In addition to this, we used a dropout rate of 0.25, two hidden layers with size 64 in each residual connection, and used Polyak averaging (Polyak & Juditsky, 1992) with a decay of 0.999 for evaluation.

## A.3 TIME SERIES

For our time series experiments, we tested the performance of flow recalibration on the UCI dataset "electricity". We filtered the data down to observations after 2014-01-01. We split the data by date, where the first 72% of dates are in the train set, the next 18% are in validation set, and the last 10% are in the test set.

The data is given in 15 minute increments, and so similarly to Deep AR (Salinas et al., 2017), we averaged the values per hour to have hourly data. Since we were trying to analyze the effect of correlated targets, we predicted two time steps conditioned on 48 hours of data using an autoregressive model.

We included three numeric covariates: the time step within the sampled time series, the hour of the day, and the day of the week. Each of the covariates were normalized to have zero mean and one standard deviation. In addition to this, since there are 370 different types of time series, we also added an embedding layer from 370 to an embedding dimension of 20, same as that used in Deep AR (Salinas et al., 2017). We concatenated this vector with the observations and covariates, leading to a 27-dimensional vector.

Since all the observations are positive, we model the target with a conditional Gamma distribution (i.e. amortized inference with Gamma distribution instead of a Gaussian). Similar to Deep AR (Salinas et al., 2017), we normalized each time series $\{z_{i,t}\}_{t=0}^{T-1}$ by $v_i = 1 + 1/T \sum_{t=0}^{T-1} z_{i,t}$. We account for the scaling in the output distribution, i.e. $\text{Gamma}(k, \theta * v_i)$.

The model we used was a 1 layer LSTM, and then two linear layers with a ReLU and Dropout (Srivastava et al., 2014) in between were applied to the hidden representation. We set the dropout rate to 0.5 and used a hidden size of 48 everywhere. For evaluation, we used Polyak averaging (Polyak & Juditsky, 1992) with a decay of 0.999.

For recalibration with flows, we trained on the CDFs for both time steps with a flow with the same architecture as that used in Appendix A.2.

## B EXPERIMENTAL RESULTS

### B.1 CALIBRATION

In Table 2, we compare Conditional Flows against multiple other approaches for estimating uncertainty, and we measure performance using Prediction Interval Coverage Probability (PICP), Mean Squared Error (MSE), and Calibration Error.

For each model, we trained the model on 20 different seeds where each seed randomly splits the data in 72% train, 18% validation, and 10% test. We picked the best hyperparameter by choosing the hyperparameters that gave the best average validation loss across the 20 seeds where the loss used is the loss defined by the training procedure for the model, e.g. for Conditional Gaussian, we used log likelihood; for LightGBM, we used MSE; for QualityDriven, we used the Quality Driven Loss, etc.

Table 2: Results on percentage of data captured from 0.025 to 0.975 quantiles (PICP), MSE, and calibration error with 100 buckets. The results are from minimizing the average validation loss across 20 seeds. The optimal value of PICP is 0.95, and the optimal value for MSE and Calib is 0.00.

| | bostonHousing | | | concrete | | |
|---|---|---|---|---|---|---|
| | PICP | MSE | Calib | PICP | MSE | Calib |
| ConditionalFlow | $0.93 \pm 0.04$ | $0.13 \pm 0.06$ | 0.06 | $0.92 \pm 0.03$ | $0.09 \pm 0.03$ | 0.01 |
| ConditionalGaussian | $0.93 \pm 0.03$ | $0.14 \pm 0.07$ | 0.13 | $0.91 \pm 0.02$ | $0.09 \pm 0.03$ | 0.06 |
| BayesianRidgeRegression | $0.95 \pm 0.03$ | $0.28 \pm 0.08$ | 0.34 | $0.95 \pm 0.03$ | $0.40 \pm 0.07$ | 0.01 |
| MC Dropout | $0.50 \pm 0.08$ | $0.11 \pm 0.05$ | 2.99 | $0.69 \pm 0.05$ | $0.07 \pm 0.03$ | 0.89 |
| SQR | $0.73 \pm 0.08$ | $0.11 \pm 0.05$ | 0.48 | $0.74 \pm 0.04$ | $0.07 \pm 0.03$ | 0.47 |
| ConditionalQuantile | $0.92 \pm 0.05$ | N.A. | N.A. | $0.93 \pm 0.03$ | N.A. | N.A. |
| QualityDriven | $0.83 \pm 0.06$ | N.A. | N.A. | $0.83 \pm 0.04$ | N.A. | N.A. |
| LightGBM | $0.04 \pm 0.03$ | $0.12 \pm 0.06$ | 7.74 | $0.63 \pm 0.07$ | $0.07 \pm 0.02$ | 1.25 |

| | energy | | | kin8nm | | |
|---|---|---|---|---|---|---|
| | PICP | MSE | Calib | PICP | MSE | Calib |
| ConditionalFlow | $0.96 \pm 0.02$ | $0.0022 \pm 0.0006$ | 0.12 | $0.952 \pm 0.006$ | $0.081 \pm 0.013$ | 0.01 |
| ConditionalGaussian | $0.97 \pm 0.03$ | $0.0030 \pm 0.0009$ | 0.26 | $0.947 \pm 0.007$ | $0.071 \pm 0.005$ | $< 0.01$ |
| BayesianRidgeRegression | $0.89 \pm 0.04$ | $0.09 \pm 0.02$ | 0.24 | $0.951 \pm 0.007$ | $0.59 \pm 0.02$ | 0.09 |
| MC Dropout | $0.97 \pm 0.02$ | $0.0026 \pm 0.0010$ | 0.17 | $0.69 \pm 0.02$ | $0.069 \pm 0.005$ | 1.14 |
| SQR | $0.97 \pm 0.02$ | $0.0028 \pm 0.0012$ | 0.37 | $0.892 \pm 0.010$ | $0.072 \pm 0.005$ | 0.01 |
| ConditionalQuantile | $0.98 \pm 0.02$ | N.A. | N.A. | $0.943 \pm 0.011$ | N.A. | N.A. |
| QualityDriven | $0.96 \pm 0.04$ | N.A. | N.A. | $0.88 \pm 0.02$ | N.A. | N.A. |
| LightGBM | $0.24 \pm 0.05$ | $0.0013 \pm 0.0006$ | 5.18 | $0.205 \pm 0.010$ | $0.190 \pm 0.011$ | 5.59 |

| | naval-propulsion-plant | | | power-plant | | |
|---|---|---|---|---|---|---|
| | PICP | MSE | Calib | PICP | MSE | Calib |
| ConditionalFlow | $0.98 \pm 0.03$ | $0.009 \pm 0.004$ | 0.45 | $0.952 \pm 0.007$ | $0.048 \pm 0.007$ | 0.01 |
| ConditionalGaussian | $0.9995 \pm 0.0007$ | $0.012 \pm 0.006$ | 2.35 | $0.947 \pm 0.010$ | $0.048 \pm 0.006$ | $< 0.01$ |
| BayesianRidgeRegression | $1.0 \pm 0.0$ | $0.055 \pm 0.004$ | 8.03 | $0.964 \pm 0.007$ | $0.072 \pm 0.007$ | 0.01 |
| MC Dropout | $0.98 \pm 0.02$ | $0.005 \pm 0.002$ | 0.51 | $0.59 \pm 0.02$ | $0.046 \pm 0.007$ | 1.88 |
| SQR | $0.998 \pm 0.002$ | $0.0052 \pm 0.0012$ | 1.44 | $0.93 \pm 0.02$ | $0.048 \pm 0.007$ | 0.02 |
| ConditionalQuantile | $0.97 \pm 0.02$ | N.A. | N.A. | $0.937 \pm 0.014$ | N.A. | N.A. |
| QualityDriven | $0.949 \pm 0.010$ | N.A. | N.A. | $0.89 \pm 0.02$ | N.A. | N.A. |
| LightGBM | $0.714 \pm 0.011$ | $0.0056 \pm 0.0009$ | 0.79 | $0.17 \pm 0.02$ | $0.035 \pm 0.007$ | 5.95 |

| | wine-quality-red | | | yacht | | |
|---|---|---|---|---|---|---|
| | PICP | MSE | Calib | PICP | MSE | Calib |
| ConditionalFlow | $0.90 \pm 0.02$ | $0.6 \pm 0.2$ | 0.04 | $0.95 \pm 0.04$ | $0.004 \pm 0.003$ | 0.27 |
| ConditionalGaussian | $0.95 \pm 0.02$ | $0.59 \pm 0.09$ | 0.22 | $0.99 \pm 0.02$ | $0.004 \pm 0.002$ | 0.62 |
| BayesianRidgeRegression | $0.95 \pm 0.02$ | $0.63 \pm 0.11$ | 0.05 | $0.94 \pm 0.03$ | $0.35 \pm 0.10$ | 0.25 |
| MC Dropout | $0.35 \pm 0.04$ | $0.57 \pm 0.09$ | 4.01 | $0.97 \pm 0.04$ | $0.004 \pm 0.002$ | 0.21 |
| SQR | $0.81 \pm 0.09$ | $0.59 \pm 0.10$ | 0.15 | $0.96 \pm 0.04$ | $0.004 \pm 0.004$ | 0.49 |
| ConditionalQuantile | $0.95 \pm 0.02$ | N.A. | N.A. | $0.99 \pm 0.02$ | N.A. | N.A. |
| QualityDriven | $0.84 \pm 0.03$ | N.A. | N.A. | $0.97 \pm 0.04$ | N.A. | N.A. |
| LightGBM | $0.40 \pm 0.05$ | $0.55 \pm 0.10$ | 2.93 | $0.45 \pm 0.08$ | $0.003 \pm 0.003$ | 3.04 |

Table 3: Calibration error (with 100 buckets) before and after recalibration.

| | BayesianRidgeRegression | | | ConditionalGaussian | | | ConditionalFlow | | |
| --- | --- | --- | --- | --- | --- | --- | --- | --- | --- |
| | Before | Flow | Iso | Before | Flow | Iso | Before | Flow | Iso |
| bostonHousing | 0.34 | 0.02 | 0.02 | 0.13 | 0.04 | 0.02 | 0.06 | 0.02 | 0.01 |
| concrete | 0.01 | 0.01 | 0.01 | 0.06 | 0.02 | 0.02 | 0.01 | 0.02 | 0.03 |
| energy | 0.24 | 0.02 | 0.01 | 0.26 | $< 0.01$ | 0.01 | 0.12 | 0.01 | 0.01 |
| kin8nm | 0.09 | 0.01 | $< 0.01$ | $< 0.01$ | $< 0.01$ | $< 0.01$ | 0.01 | $< 0.01$ | $< 0.01$ |
| naval-propulsion-plant | 8.03 | 0.01 | $< 0.01$ | 2.35 | $< 0.01$ | $< 0.01$ | 0.45 | $< 0.01$ | $< 0.01$ |
| power-plant | 0.01 | $< 0.01$ | $< 0.01$ | $< 0.01$ | $< 0.01$ | $< 0.01$ | 0.01 | $< 0.01$ | $< 0.01$ |
| wine-quality-red | 0.05 | 0.02 | $< 0.01$ | 0.22 | 0.04 | 0.02 | 0.04 | 0.01 | 0.01 |
| yacht | 0.25 | 0.05 | 0.07 | 0.62 | 0.03 | 0.03 | 0.27 | 0.02 | 0.03 |

Table 4: MSE performance before and after recalibration. We do not show the MSE for isotonic regression as the mean cannot be computed after recalibrating with isotonic regression.

| | BayesianRidgeRegression | | ConditionalGaussian | |
| --- | --- | --- | --- | --- |
| | Before | Flow | Before | Flow |
| bostonHousing | $0.28 \pm 0.08$ | $0.28 \pm 0.08$ | $0.14 \pm 0.07$ | $0.14 \pm 0.07$ |
| concrete | $0.40 \pm 0.07$ | $0.40 \pm 0.07$ | $0.09 \pm 0.03$ | $0.09 \pm 0.03$ |
| energy | $0.09 \pm 0.02$ | $0.09 \pm 0.02$ | $0.0030 \pm 0.0009$ | $0.0032 \pm 0.0011$ |
| kin8nm | $0.59 \pm 0.02$ | $0.59 \pm 0.02$ | $0.071 \pm 0.005$ | $0.072 \pm 0.005$ |
| naval-propulsion-plant | $0.055 \pm 0.004$ | $0.48 \pm 0.02$ | $0.012 \pm 0.006$ | $0.012 \pm 0.007$ |
| power-plant | $0.072 \pm 0.007$ | $0.072 \pm 0.007$ | $0.048 \pm 0.006$ | $0.049 \pm 0.006$ |
| wine-quality-red | $0.63 \pm 0.11$ | $0.63 \pm 0.11$ | $0.59 \pm 0.09$ | $0.59 \pm 0.10$ |
| yacht | $0.35 \pm 0.10$ | $0.36 \pm 0.10$ | $0.004 \pm 0.002$ | $0.004 \pm 0.003$ |

| | ConditionalFlow | |
| --- | --- | --- |
| | Before | Flow |
| bostonHousing | $0.13 \pm 0.06$ | $0.13 \pm 0.06$ |
| concrete | $0.09 \pm 0.03$ | $0.09 \pm 0.03$ |
| energy | $0.0022 \pm 0.0006$ | $0.0027 \pm 0.0007$ |
| kin8nm | $0.081 \pm 0.013$ | $0.081 \pm 0.013$ |
| naval-propulsion-plant | $0.009 \pm 0.004$ | $0.016 \pm 0.012$ |
| power-plant | $0.048 \pm 0.007$ | $0.048 \pm 0.007$ |
| wine-quality-red | $0.6 \pm 0.2$ | $0.64 \pm 0.15$ |
| yacht | $0.004 \pm 0.003$ | $0.008 \pm 0.012$ |

Reported in Table 2 are the mean and standard deviation of PICP and MSE across the 20 seeds. Due to the limited data, for calibration error with 100 bins, we computed it over the CDFs from all 20 seeds. We included PICP (the percentage of true observations between the quantiles 0.025 and 0.975) to compare against alternative approaches where we prespecify which quantiles we need.

## B.2 RECALIBRATION

From the above section, we recalibrate the best Bayesian Ridge Regression, Conditional Gaussian, and Conditional Flow. In Table 3, we compare the calibration error before recalibration, after recalibration with isotonic regression, and after recalibration with normalizing flows.

As we can see, across all 8 UCI datasets, the calibration errors after recalibrating with normalizing flows are comparable to those from isotonic regression. More interestingly though, improvement in calibration error can be detrimental to mean performance.

