# OpenReview forum: "Normalizing Flows for Calibration and Recalibration"
_ICLR.cc/2021/Conference — Reject_

### Official Review · AnonReviewer2 · 2020-10-28
**seems like a good paper, but I am not an expert**

**Rating:** 7
**Confidence:** 1

**Review:**

The paper proposes to use normalizing flows to improve estimates of aleatoric uncertainty in regression tasks. First, the paper suggests that since normalizing flows can improve the flexibility of output distribution, they can be used to mitigate issues of underfitting. Second, the paper proposes an approach that uses normalizing flows for recalibration, which allows people to still query the model’s statistical properties. The authors also introduce a plot that is useful for diagnosing calibration issues.

The paper’s approach of applying normalizing flows in recalibration seems that it could be useful to the community. The supporting experimental results look reasonable. In addition, the presentation of the paper looks nice, the experiments are well documented, and the diagnosing plots seem like a helpful tool. Given these plus points, I would recommend acceptance.

One suggestion is that it feels quite obvious that since flows can represent flexible distributions, using them to model aleatoric uncertainty can reduce underfitting issues. I am not sure if it is worth stating in the paper. It seems that the interesting part of the paper is recalibration, so perhaps it might be better to focus more on that.

Questions for the authors:
- I wonder whether using flows to recalibrate is susceptible to overfitting.
- The paper focuses on aleatoric uncertainty. How can the approach proposed in the paper be combined with approaches for addressing epistemic uncertainty?

---

> ### Author Response · Authors · 2020-11-17
> **Response to Reviewer 2**
>
> 1. *I wonder whether using flows to recalibrate is susceptible to overfitting.*
>
> Flows can also be susceptible to overfitting; more often if you condition the flow on many variables. For similar reasons as to why isotonic regression, while being non-parametric, does not overfit, normalizing flows fit only on one dimensional CDFs will also not overfit. However, the more variables you condition on or the more variables you train on jointly, the more likely the flow can overfit.  In the extreme case, if you give all the features to the recalibration step, the model is doing exactly what was done during the calibration step, i.e. equally susceptible as the calibration step, if not more so due to the smaller validation set.
>
> 2. *The paper focuses on aleatoric uncertainty. How can the approach proposed in the paper be combined with approaches for addressing epistemic uncertainty?*
>
> Though this technique handles aleatoric uncertainty, one interpretation (though not proven) is that recalibration is handling both any miscalibration because of misspecification of the aleatoric uncertainty but also any overfitting due to limited data which lies closer to epistemic uncertainty. We leave to future work a way to decompose the results of recalibration into these two uncertainties.

---

### Official Review · AnonReviewer1 · 2020-10-28
**Interesting model but with unconvincing improvements over recalibration baseline**

**Rating:** 5
**Confidence:** 3

**Review:**

This paper proposes a method for calibrating uncertainty estimates for regression models. It builds off a method proposed by Kuleshov et al (2018). The newly proposed method has the following steps:
  1) Train a conditional density model for a regression outcome y given an input x on training data. The authors use conditional normalizing flows for this task.
  2) For each set of points (x_t, y_t) in a validation set, pass an input x_t to the model in step 1 to learn an induced CDF for the output F_t(). Calculate F_t(y_t), i.e. the induced CDF evaluated at the actual output, and use another flow-based density model to learn the distribution of the CDF values. If the model in step 1 is perfectly calibrated, this density should be uniform, but in practice it seldom is.
  3) For future data points, the composition of densities in steps 1) and 2) provide a new, uncertainty-calibrated density.

The overall problem area is interesting, important, and underexplored. There has been a lot of work on calibration estimates for classification tasks, but there are less methods for regression. Using flows here is a cool idea. Any exact density model can be used for the method above, and normalizing flows are a good solution because they are monotonic by construction.

That being said, after reading the paper, I'm not convinced that the proposed method is a significant improvement over the method by Kuleshov et al. that it's building off of. Kuleshov et al. proposes a similar 2-step process, but instead of explicitly learning a distribution over the induced CDF values, it uses an isotonic regression to calibrate the CDF. Looking at the experimental results, it seems that the isotonic recalibration performs on par with the flow recalibration in terms of test set calibration error.

Put another way, why should someone use flow-based recalibration instead of isotonic recalibration? A possible answer mentioned in the paper is that flow-based recalibration can be used to compute distribution statistics, such as the mean, while isotonic recalibration cannot compute these statistics. (Side question: why can't the isotonic recalibration be used to compute the mean? It seems like isotonic recalibration explicitly transforms an inverse-CDF to another inverse-CDF. Can't distribution statistics be imputed from this transformed inverse-CDF?).

Even though flow recalibration can compute distribution statistics, the MSE never improves after flow recalibration; in some instances, it gets worse. What is the benefit of having distribution statistics? On the one hand, the worsened MSE might be expected behavior. Is uncertainty calibration expected to behave like a regularizer? If so, that should be stated and discussed in the paper. If not, then of course we shouldn't expect improvements in MSE after recalibration, because we're changing the model that had the best training-set performance. This could explain the results we see. In any case, there should be some justification in the paper for a) why computing distribution statistics is important, b) whether we should expect recalibration to behave like a regularizer, c) a discussion about the tradeoff between calibration performance and model error, and d) an illustration of scenarios where distribution statistics are crucial [and e), why the Kuleshov et al method can't be used to calculate test error].

Additionally, the paper proposes a way to visualize recalibration results. To be honest, I found the CDF performance plot confusing and hard to interpret. How should we interpret the x-axis (are predictions standardized, and if not, what units are they in)? I found the standard qq-plot-like calibration graph a lot more interpretable. What does the new visualization answer? I think the new visualization should be better explained (it also didn't help that the legend in Figure 5 blocked the middle of the graph).

Overall, I think the paper proposes an interesting model, but it doesn't adequately justify when/why the model should be used over the existing method. I think there could definitely be scenarios where it is useful -- I just don't think the paper has adequately and convincingly illustrated them.

Pros:
- Interesting and underexplored problem area
- Thorough experiments
- Normalizing flows are an interesting and new model for this problem

Cons:
- Doesn't justify meaningful ways the method is different from existing methods
- Visualizations are confusing

---

> ### Author Response · Authors · 2020-11-17
> **Response to Reviewer 1**
>
> 1.  *a) why computing distribution statistics is important [,...] d) an illustration of scenarios where distribution statistics are crucial*
>
> In the situation in which the only point estimate one cares about is the mean, we would not need to worry about the uncertainty metrics or recalibration.  If we are considering recalibration, then we are interested in a model that is, at the very least, capable of modeling any arbitrary quantile of a distribution.  When using a full distribution we arrive at a model that, when properly calibrated, gives access to not only arbitrary quantiles, but other statistics such as the mean, variance, and other higher moments.  In the multivariate case it also allows us to access correlations.  It is important to note that using a full distribution achieves this with a single model whereas using loss functions that only model a single statistic would require one model per statistic of interest.  In addition, it would preclude any downstream likelihood use cases (e.g., anomaly detection) as well as anything that makes use of the generative model (e.g., samples).
>
>
> 2. *b) whether we should expect recalibration to behave like a regularizer*
>
> If we view the test error as MSE, then one could view calibration/recalibration as a regularizer since the model capacity has to be used to both accurately learn the mean as well as all other higher order moments of the distribution.
>
> 3. *e) why the Kuleshov et al method can't be used to calculate test error*
>
> The reason isotonic regression cannot be used to compute test MSE is because the model is CDF to CDF, and so either you can try to sample from it but you would have to find the inverse of the model or you can try to compute the log likelihood which you would do by trying to understand the change of variables implied by the nonparametric method. In both cases, the methods aren’t impossible but are not naturally incorporated into isotonic regression, discounting the fact that isotonic regression also cannot handle conditional variables.
>
> 4. *What does the new visualization answer?*
>
> The goal of the CDF performance plot is to show the calibration performance across different subsets of the data (where in the original example, we were binning by the model’s mean prediction). In order to simplify the presentation and focus more on the usage of normalizing flows, we show the same content via CDF Q-Q Plots.
>
> 5. *Doesn't justify meaningful ways the method is different from existing methods*
>
> Thanks for the feedback, we have updated the introduction and conclusion to better justify how normalizing flows differ from isotonic regression. The main benefits here are that normalizing flows can be trivially extended to being conditional and multivariate. Conditional recalibration was mentioned in Kuleshov et al. (2018), but the technique there is using a ConditionalGaussian, instead of something non-parametric like isotonic regression. We specifically use QuAR Flows as opposed to other normalizing flows and conditional distributions as they are highly flexible (making it comparable to isotonic regression) and computationally efficient.

---

> > ### Comment · AnonReviewer1 · 2020-11-23
> > **Response**
> >
> > Thank you for the response. The role of the proposed method -- and how it differs with that introduced by Kuleshov et al -- is more clear to me now. That being said, the experimental results don't offer a very compelling story for improvement. My advice for the next revision would be to find tasks where it is vital to compute arbitrary quantiles of interest. The ability to compute arbitrary quantiles appears to be the main advantage of the proposed model, but the experiments haven't found a compelling, practical use case.

---

### Official Review · AnonReviewer4 · 2020-10-29
**Difficult to follow due to missing details**

**Rating:** 4
**Confidence:** 4

**Review:**

The paper proposes a normalizing flow approach for calibrated predictions in regression tasks. Experimental results on toy and UCI datasets demonstrate the proposed method improves model calibration.

It is not clear what the technical contributions are. Also, most of the important details are missing.

**Strengths**
- The reviewer appreciates the effort towards improved calibration models; important for reliable predictions

**Weaknesses**

*Lack of clarity*:
- The paper was difficult to follow, it omits several crucial details necessary to understand the proposed method.

Below are some general questions or suggestions:
1) While the paper's principal focus is on calibration and recalibration, it is unclear why there are claims to address aleatoric uncertainty
2) The concept of recalibration is introduced in Section 2 as a classification problem. However, in Section 4, the focus is on regression with normalizing flows
3)  In Figures 1 and 2, what is $X, c, W_1, W_2$?
4) Figure 2, add labels to x-axis and y-axis
5) Improve caption quality across all Figures and tables

*CDF performance plot*:
- The paper claims the CDF performance plot is one of the main contributions, yet it is difficult to interpret or follow. Why are calibrated CDFs uniformly distributed?
1) What is $\sigma$?
2) What is **pull**?
3) What is $\psi$

*Recalibration Normalizing flows*:
- What are the complete formulations of the likelihoods optimized in 1) and 2)?
-  How is normalizing flow extended to multivariate calibration?

*Weak experiments*:
- How is the *Calib*  metric computed?
- The experiments are on toy data and small UCI datasets. Additional large scale image datasets or regression tasks would strengthen the submission.

---

> ### Author Response · Authors · 2020-11-17
> **Response to Reviewer 4**
>
> 1. *While the paper's principal focus is on calibration and recalibration, it is unclear why there are claims to address aleatoric uncertainty*
>
> By definition, a well calibrated conditional density model will capture the aleatoric uncertainty.
>
> 2. *The concept of recalibration is introduced in Section 2 as a classification problem. However, in Section 4, the focus is on regression with normalizing flows*
>
> Research focus, thus far, has largely focused on classification.  While the paper introduces the concept of recalibration both within the context of classification and regression, the focus of the paper is on improving the as-of-yet less explored case of regression.
>
> 3. *In Figures 1 and 2, what is X,c,W1,W2?*
>
> X is the input vector, c is an arbitrary vector that the recalibration is (optionally) conditioned on, W1 and W2 are the weight matrices that parameterize the residual layer.  This section has been rewritten for clarity.
>
> 4. *What are the complete formulations of the likelihoods optimized in 1) and 2)?*
>
> The likelihoods specified in 1) and 2) are from the individual model that is being recalibrated. As noted in the paper, the only restriction that is imposed by using normalizing flows for recalibration is that we assume the original model outputs a full distribution that has closed-form likelihoods.  This could be as simple as a Gaussian and as complex as another normalizing flow.
>
> 5. *How is normalizing flow extended to multivariate calibration?*
>
> Section 4.3 addresses the extension to multivariate calibration.  A complete example has been added for clarity.
>
> 6. *How is the Calib metric computed?*
>
> Thank you for the feedback. We added more detail in the caption to explain how we computed the calibration error on the UCI datasets.
>
> 7. *The experiments are on toy data and small UCI datasets. Additional large scale image datasets or regression tasks would strengthen the submission.*
>
> The goal of our paper is to show that there are failure points that arise from isotonic regression that normalizing flows are better suited to fixing.  Namely, multivariate recalibration and the inability to calculate arbitrary statistics on the recalibrated distribution apart from sampling.

---

### Official Review · AnonReviewer3 · 2020-10-30
**Not sure what is novel here.**

**Rating:** 3
**Confidence:** 5

**Review:**

Summary:
The authors propose an approach to calibrate conditional distribution estimation models. The approach uses normalizing flows to transform an existing model's predictions into a prediction that better matches the empirical quantiles to the theoretical quantiles. After this remapping procedure, the authors introduce a new plot to visualize calibration. Empirical benchmarks are run on a suite of UCI datasets.

Review:

I'm not sure what is really that interesting here. My high-level problems:

- The remapping that the authors propose is just using a normalizing flow with a simple quantile calibration. Why do we need normalizing flows for this at all? Any model can be recalibrated using any other model here. Is there some special reason for normalizing flows here?

- The new plot introduced is more confusing than illuminating. I really didn't follow it at all. It is very crowded and takes a lot of inspection to understand what is going on. I suspect all of this could have been visualized much clearer by using a handful of simpler plots that are straightforward.

- The benchmarks do not really compare against very competitive models. The authors choose to use a model that was only pushed to the arxiv a month ago as the baseline. Why? Then the alternative choices are strawmen: a Bayesian linear regression model, a variational dropout model, etc. There is a wealth of conditional distribution estimation literature with companion code publicly available on github (NADE, MAF, MDNs, etc). Why not use those?

- Does this really help us do anything new? Is this just "my model is 0.1% better on 8 UCI datasets than 4 other models"? Seems like a pretty uninspiring result if that's the idea.

Overall, I just don't know what to see as the big contribution here. The paper feels a little rushed and could use a slower, more methodical pace where the authors carefully think through the contribution(s) and why they're necessary. A more thorough comparison to related work is also called for.

---

> ### Author Response · Authors · 2020-11-17
> **Response to Reviewer 3**
>
> 1. *The remapping that the authors propose is just using a normalizing flow with a simple quantile calibration. Why do we need normalizing flows for this at all? Any model can be recalibrated using any other model here. Is there some special reason for normalizing flows here?*
>
> Normalizing flows allow us to recast the recalibration problem as a maximum likelihood problem.  Due to their monotonicity by construction, we are able to retain a tractable, explicit density in the recalibrated model, from which we can then recover any statistic.
>
> 2. *The new plot introduced is more confusing than illuminating. I really didn't follow it at all. It is very crowded and takes a lot of inspection to understand what is going on. I suspect all of this could have been visualized much clearer by using a handful of simpler plots that are straightforward.*
>
> Thank you for the feedback.  In order to maintain the focus of the paper on the usage of normalizing flows for recalibration we have presented the results in a more familiar, albeit less complete, manner with multiple QQ plots.
>
> 3. *The benchmarks do not really compare against very competitive models. The authors choose to use a model that was only pushed to the arxiv a month ago as the baseline. Why? Then the alternative choices are strawmen: a Bayesian linear regression model, a variational dropout model, etc. There is a wealth of conditional distribution estimation literature with companion code publicly available on github (NADE, MAF, MDNs, etc). Why not use those?*
>
> The conditional flow model used (QuAR) was chosen as it is an evolution of residual flows that retains most, if not all, of the performance while improving on the computational efficiency.  The alternatives are chosen as examples of other models that are used to attempt to model the aleatoric uncertainty in regression problems.
>
> For other deep learning approaches to modeling conditional distributions such as NADE and MAF, in the one dimensional case, they reduce down to the ConditionalGaussian that we used. Some techniques do replace the output distribution with mixtures or other analytical distributions; however QuAR Flows were shown to be able to handle mixtures as well as interesting shapes like a Uniform (where other techniques would struggle unless it was assumed that the output is uniformly distributed).
>
> 4. *Does this really help us do anything new? Is this just "my model is 0.1% better on 8 UCI datasets than 4 other models"? Seems like a pretty uninspiring result if that's the idea.*
>
> The intent is not to show that we can recalibrate models better on the UCI datasets, but rather that normalizing flows have competitive performance in the space of problems that isotonic regression can solve, while also maintaining a tractable density from which we can calculate any statistic (in direct contrast to isotonic regression).  Most importantly, isotonic regression is not appropriate for the recalibration of multivariate regression models as it is unable to account for the correlations between the individual variables.  Normalizing flows give us a path to recalibrate a multivariate distribution properly.
>
> In order to better highlight this novelty we have added section 5, in which we give a simple example where multivariate recalibration with isotonic regression would fail, yet recalibration with normalizing flows is successful.

---

### Author Response · Authors · 2020-11-17
**Paper Updated**

Thanks to all the reviewers for their feedback. From the feedback, we have incorporated the following changes:

1. *Contributions and Novelty*

Updated the conclusion to better summarize the contributions of our paper to recalibration.  Namely, recasting recalibration as MLE, multivariate recalibration, and the ability to calculate arbitrary statistics on the recalibrated distribution apart from sampling.

2. *CDF Performance Plot*

In order to maintain the focus of the paper on the usage of normalizing flows for recalibration, we have presented the results in a more familiar, albeit less complete, manner with multiple QQ plots.

3. *Multivariate Recalibration*

To better explain multivariate recalibration and its usefulness, we have added another section walking through a simple case as well as a diagram showing the pipeline for multivariate recalibration.

---

### Decision · Program_Chairs · 2021-01-07
**Final Decision**

**Decision:**

Reject

**Comment:**

The paper proposes to recalibrate predictive models by fitting a
normalizing flow on top of the predictive model on a held out validation
set using side information. At a high level this idea has some potential,
especially in the multivariate setting, but there are several directions for
improvement:

- Comparison with a broader set of baselines as suggested by the reviewers


- Clarity on why recalibrate with a normalizing flow especially in the 1-d case


- Why not any other model with explicit density? Are there other important desiderata?


- A motivating experiment that makes the potential value clear